# Using GPT-3 to Build a Lexicon of Drugs of Abuse Synonyms for Social Media Pharmacovigilance

**DOI:** 10.3390/biom13020387

**Published:** 2023-02-18

**Authors:** Kristy A. Carpenter, Russ B. Altman

**Affiliations:** 1Department of Biomedical Data Science, Stanford University, Stanford, CA 94305, USA; 2Departments of Bioengineering, Genetics, and Medicine, Stanford University, Stanford, CA 94305, USA

**Keywords:** large language models, pharmacovigilance, social media, drugs of abuse

## Abstract

Drug abuse is a serious problem in the United States, with over 90,000 drug overdose deaths nationally in 2020. A key step in combating drug abuse is detecting, monitoring, and characterizing its trends over time and location, also known as pharmacovigilance. While federal reporting systems accomplish this to a degree, they often have high latency and incomplete coverage. Social-media-based pharmacovigilance has zero latency, is easily accessible and unfiltered, and benefits from drug users being willing to share their experiences online pseudo-anonymously. However, unlike highly structured official data sources, social media text is rife with misspellings and slang, making automated analysis difficult. Generative Pretrained Transformer 3 (GPT-3) is a large autoregressive language model specialized for few-shot learning that was trained on text from the entire internet. We demonstrate that GPT-3 can be used to generate slang and common misspellings of terms for drugs of abuse. We repeatedly queried GPT-3 for synonyms of drugs of abuse and filtered the generated terms using automated Google searches and cross-references to known drug names. When generated terms for alprazolam were manually labeled, we found that our method produced 269 synonyms for alprazolam, 221 of which were new discoveries not included in an existing drug lexicon for social media. We repeated this process for 98 drugs of abuse, of which 22 are widely-discussed drugs of abuse, building a lexicon of colloquial drug synonyms that can be used for pharmacovigilance on social media.

## 1. Introduction

The opioid epidemic is a growing crisis, driving a drastic rise in deaths attributed to drug overdose over the past several years in the United States [1,2]. Of the nearly 92,000 overdose deaths in 2020, over 56,000 involved synthetic opioids such as fentanyl [3]. It is imperative for researchers to understand the past, present, and future of drug abuse in order to combat this national emergency.

Pharmacovigilance is the detection, assessment, and analysis of the usage and effects of drugs [4]. Monitoring trends in the opioid epidemic and the abuse of other drugs is a critical first step in reducing the number of deaths from drug overdoses [5]. Several international and national agencies, such as the World Health Organization (WHO), the European Medicines Agency (EMA), the U.S. Food and Drug Administration (FDA), the U.S. Centers for Disease Control and Prevention (CDC), the U.S. National Institutes of Health (NIH), the U.S. Drug Enforcement Administration (DEA), and the U.S. Department of Health and Human Services (HHS), survey and monitor drug use and effects. Notable pharmacovigilance systems from these agencies include VigiBase, EudraVigilance, the FDA Adverse Event Reporting System (FAERS), the National Health and Nutrition Examination Survey (NHANES), the National Drug Early Warning System (NDEWS), the National Forensic Laboratory Information System (NFLIS), and the National Survey on Drug Use and Health (NSDUH). Concern about growing opioid abuse has driven numerous analyses on opiates’ pharmacovigilance data from these systems [6,7,8,9,10,11,12,13]. In addition to analysis of statistics and metrics from reporting systems, pharmacovigilance studies have also been conducted using text mining and natural language processing (NLP) on free text notes in electronic health records (EHRs) [14,15,16,17,18,19].

There has been an increased interest over the past decade in using social media for NLP-based pharmacovigilance. While official pharmacovigilance surveys may have a latency of months to years to make results available, social media data can be queried nearly instantaneously. Drug users are also willing to freely post their experiences with drugs online, sharing information that they may not want to make accessible to federal agencies, making social media a valuable resource for surveillance of illicit drug use [20]. The pseudo-anonymity offered by various social media platforms facilitates this openness. Social media pharmacovigilance is believed to have first appeared in 2010 [21] and has gained traction since, with most studies using the social media platforms Twitter [22,23,24,25,26,27,28,29,30], Facebook [28,29,30], and Reddit [31,32,33] for tasks such as adverse drug reaction extraction and off-label drug usage analysis.

While social media holds promise for improving pharmacovigilance efforts, it also brings unique challenges. Social media data are fundamentally different from that of the FDA, WHO, CDC, NIH, or even clinical notes in an EHR in that it is by nature casual and unstandardized, and therefore rife with misspellings and slang. There is abundant missingness, as not all posts will contain geographic or demographic information. Misinformation is rampant on social media, with the ease of posting and incentivization of viral content leading to the easy spread of rumors, conspiracy theories, and misleading interpretations of scientific results, both intentionally and unintentionally [34,35]. In many cases, it is impossible to verify the validity of any information posted. The contents of social media are also heavily influenced by politics, current events, and pop culture. As such, analyses of social media could be considered “unscientific”, or at least in violation of the traditional standards of epidemiological studies. There is much improvement to be desired from social media pharmacovigilance efforts [36], which are still very much in their infancy. However, despite these limitations, multiple meta-analyses and qualitative reviews have found that social media pharmacovigilance efforts are able to extract some meaningful signal pertaining to drug use and effects [37,38,39].

One method that addresses some of the outstanding problems of social media pharmacovigilance is RedMed [40], a word embedding model based on continuous bag-of-words modeling [41] and trained on archived comments on health- and drug-related Reddit forums. After training the model to cluster similar terms, RedMed can discover candidate terms with significant cosine similarity to an index term from DrugBank [42] and subsequently verify those terms with filters related to edit distance, phonetics, pill impressions, and Google search results. RedMed produced a lexicon of drug synonyms that included misspellings and slang terms, enabling better retrieval of pharmacovigilance-relevant text from social media sources; it was subsequently used for quantification of adverse drug reaction severity [32].

We propose to extend RedMed without training a new embedding model by using pre-trained large language models—specifically, Generative Pretrained Transformer 3 (GPT-3) [43]. GPT-3 is the third installation of a generative pre-trained transformer from the company OpenAI that has been trained on the entire internet. It is an autoregressive language model of unprecedented size, with 175 billion parameters. Generally, language models are probability distributions over sequences (typically of words) that can identify if a given sequence is likely or generates likely sequences. Transformers are a machine learning architecture that is built around the attention mechanism [44], and have sparked great advances in language modeling. GPT-3 garnered much discussion upon its release in 2020 due to its performance in few-shot learning; given only a few examples, it is able to produce desired text output that closely resembles real human writing. Typical examples of GPT-3 tasks are question-answering, story completion, translation, and summarization. Researchers have also explored using GPT-3 in a medical context, on tasks such as EHR summarization or supporting a medical chatbot, but no such models have been deemed ready for deployment in the clinic [45,46,47]. The reception of GPT-3 has not all been positive, and due to its training on the entire internet, it is prone to generating text that perpetuates harmful stereotypes or promotes dangerous activity [43,48,49]. OpenAI has made GPT-3 available as an application programming interface (API), allowing researchers to leverage its capabilities without needing to train a massive language model themselves.

We argue that GPT-3 is valuable for social media pharmacovigilance as it is able to generate text that closely resembles common writing patterns used on the internet at large. In this work, we make the following contributions:We introduce a novel method to repeatedly query GPT-3 for drug synonyms and filter the generated terms to create a lexicon enriched for likely synonyms, all in an automated fashion. We make the code for the method publicly available to build similar lexicons, facilitating interpretable pharmacovigilance on messy, casually-written social media data that does not require training a new large machine learning model;We present a lexicon of GPT-3 synonyms for 98 drugs of abuse, including 22 widely-discussed drugs of abuse, which can be used to easily flag text likely to be related to drug abuse from a large corpus of informal language in an interpretable manner;Finally, we also demonstrate of the capabilities of GPT-3, and similar models, for practical contributions to pharmacovigilance.

Code and data are available on GitHub (https://github.com/kristycarp/gpt3-lexicon).

## 2. Materials and Methods

### 2.1. Datasets

#### 2.1.1. RedMed

We use index terms from the RedMed lexicon to provide seed terms to GPT-3. The RedMed lexicon is comprised of index terms from DrugBank, their respective associated known drug terms (AKDTs) (e.g., brand names), and their respective synonymous terms generated by a word embedding model and subsequently filtered. The terms in the RedMed lexicon data frame are organized into columns to indicate how the term was validated: because it is an AKDT (known), within close edit distance (edOne, edTwo), within close phonetic edit distance (misspellingPhon), a pill impression (pillMark), validated by Google search (google_ms, google_title, google_snippet), or present in a slang-specific database (ud_slang). To improve quality of our results, we only sample from the RedMed synonyms that are a single-word AKDT, within close edit distance, within close phonetic edit distance, and a pill impression. The choice to limit inclusion of AKDTs to only those comprised of a single word followed from the observation that many multi-word AKDTs were simply short phrases containing the seed term (e.g., for alprazolam, commonly known as xanax, the multi-word AKDTs include “started taking alprazolam”, “xanax works great”, and “quit taking xanax”). When these phrases are presented to GPT-3, more such phrases are generated, which are not useful for our task as they already contain a known drug synonym and therefore add no information. We also excluded RedMed synonyms from the google_ms, google_title, google_snippet, and ud_slang columns, as these tended to include a higher rate of false positives, and presenting GPT-3 with irrelevant examples leads to generation of more irrelevant terms.

#### 2.1.2. Drugs of Abuse

The DEA maintains a list of controlled substances, which are defined to be drugs with high potential for abuse. As of July 2022, there are 543 DEA-controlled substances [50]. We took the intersection of the 543 controlled substances and the 2997 index terms in RedMed, resulting in 131 controlled index terms. Of these, 33 contained fewer than three terms in our selected columns of RedMed; we eliminated these index terms as they did not have sufficient RedMed synonyms for GPT-3 prompt generation. This resulted in a final set of 98 controlled index terms to input into the GPT-3 query pipeline.

#### 2.1.3. Widely-Discussed Drugs of Abuse

Some of the 98 selected index terms are more widely-discussed online than others, and therefore would likely have more synonyms than less widely-discussed index terms. In order to better evaluate how our pipeline performs on these drugs, we took approximately the top 25% of index terms with respect to discussion on Reddit and created a subset of widely-discussed drugs of abuse. For each of the 98 index terms, we used Google to search for exact matches to the index term on Reddit. We choose to limit to Reddit to reduce noise and because Reddit is a popular platform for discussing drug use [40,51,52]. The cutoff for the top 25% of index terms was approximately 10,000 Reddit hits (Figure A1), so for simplicity we used 10,000 hits on Reddit webpages as the cutoff to determine if an index term should be included in the “widely-discussed” subset. Of the 98 selected index terms, 22 are “widely-discussed.” We note that this subset is only intended to demonstrate pipeline performance on drugs which are discussed more frequently on relevant discussion forums, as they are more likely downstream applications of our lexicon and pipeline than less prominent drugs.

### 2.2. External Models

#### 2.2.1. GPT-3

We accessed the GPT-3 model [43] through the OpenAI API. We used the text-davinci-002 engine for all queries.

#### 2.2.2. Google Search API

We used the Custom Search JSON API from Google’s Programmable Search Engine to automate Google searches of generated terms.

### 2.3. Terminology

Terminology coined in this manuscript (or in [40] and key to this study) is defined in Table 1.

### 2.4. Methods

#### 2.4.1. Overview of Query Pipeline

An iteration of the query pipeline begins by uniformly sampling three RedMed synonyms for the queried index drug term. We insert the index term and the sampled RedMed synonyms into a prompt template (further described below), which we provide to GPT-3 as a Completion query. Because we use an enumerative list in our prompt templates, and because GPT-3 is easily able to pick up on enumerative formatting, nearly all results returned by GPT-3 will also be formatted in an enumerative list. We automatically parse the listed results to extract the GPT-3 generated terms. We repeat this process to build a set of GPT-3 generated terms for the queried index term. We also pass the generated terms through filters described below. A schematic of this GPT-3 querying pipeline is depicted in Figure 1.

#### 2.4.2. GPT-3 Prompt Templates

We experimented with a variety of prompt templates at a small scale in a sandbox environment when constructing the format of GPT-3 queries. Examples included asking for synonymous terms with and without examples, asking for synonymous terms in a colloquial manner (using slang and misspellings in the prompt), and writing the prompt as a conversation between two drug users discussing slang terms. We observed that prompts formulated as an enumerative list most often led to GPT-3 completions that continued the list, facilitating automated parsing of generated terms. Other types of prompts (such as asking in a colloquial manner or framing the prompt as a conversation) led to responses that were too varied to easily extract sets of generated terms at scale. In addition, because GPT-3 is specialized for few-shot learning [43], we know that it works very well for a desired task when given a few examples of desired output, and saw this reflected in our small-scale prompt experiments. These observations resulted in the choice of the following prompt template:

“ways to say [index term]:

1. [RedMed synonym 1]

2. [RedMed synonym 2]

3. [RedMed synonym 3]

4.”

The hanging “4.” indicates to the model that it should continue filling in the list.

We chose to include three example synonyms in the prompt template because we observed in our small scale experiments that GPT-3 tended to complete the enumerated list until there were three or ten items in the list. Therefore, using three example synonyms often led to seven additional terms being generated, maximizing the number of generated terms when this pattern was followed. Because we did not observe a drastic change in the number of generated terms beyond this pattern, we chose to not further investigate varying the number of synonyms presented in the prompt template, though this could become an area of future work. We note that GPT-3 queries limit the number of tokens in the prompt and response combined, meaning that listing a large number of example synonyms could impact the number of terms able to be generated.

We also observed that, with this formulation, GPT-3 tends to generate the names of drugs that are different from the index term but have the same indications (e.g., generating the names of other anti-anxiety medications when prompted for alprazolam terms). We hypothesized that providing counterexamples in the prompt might reduce this phenomenon. Our prompt template with counterexamples is as follows:

“these are not synonyms for [index term]:

1. [counterexample 1]

2. [counterexample 2]

3. [counterexample 3]

4. [counterexample 4]

but these are synonyms for [index term]:

1. [RedMed synonym 1]

2. [RedMed synonym 2]

3.”

In our parameter search experiments, we used hand-picked counterexamples; for the index term of alprazolam, our counterexamples were ativan, zoloft, lexapro, and klonopin.

#### 2.4.3. GPT-3 Parameter Search

The GPT-3 query API allows for the specification of model parameters, which include temperature, frequency penalty, and presence penalty. Temperature indicates how much the model should prioritize high-likelihood answers over providing diverse answers and ranges from 0 to 1; a low temperature leads to the model prioritizing high-likelihood answers, and a high temperature leads to the model prioritizing diverse answers. The frequency penalty controls how likely the model is to generate the same tokens verbatim and ranges from −2 to 2, with more positive numbers increasing the penalty of this verbatim repetition. In this context, a token is a sequence of characters (often full words, though a word can also be comprised of multiple tokens) commonly found in the training corpus of GPT-3. GPT-3 functions by learning the statistical relationships between tokens [43]. The presence penalty controls how likely the model is to generate text about new topics and ranges from −2 to 2, with more positive numbers increasing the penalty of topic repetition. We sought to identify the model parameters, as well as the prompt template that would maximize the number of unique novel GPT-3 synonyms (UNGSes), which we define as generated terms that pass the post-query filters and are not already present in RedMed. We only want to count each unique generated term once, as generating the same term multiple times does not add new information to the lexicon. We do not want to count terms that are already RedMed synonyms because these were already known and available. We ran 1000 iterations of the query pipeline on one index term for each possible combination of the following parameter settings: temperatures of 0.0, 0.3, 0.6, and 1.0; frequency penalties of 0.0, 0.5, and 1.0; presence penalties of 0.0, 0.5, and 1.0; and the prompt templates with and without counterexamples. We chose to only investigate these settings, rather than conduct a full automated parameter sweep, due to budget constraints (both the OpenAI API and the Google Search API incur costs per query) and the rationale that a grid search of values spanning the ranges of each parameter would be sufficient to identify settings useful for downstream application. We selected alprazolam as the index term for the initial parameter sweep experiment because it is a common drug of abuse that is discussed widely online and therefore a representative example of the type of term for which we would like good performance. We confirmed the observed trends from alprazolam by additionally running 1000 iterations of the query pipeline on two more drugs on a smaller set of parameter setting combinations: temperatures of 0.0, 0.5, and 1.0; frequency penalties of 0.0 and 1.0; presence penalties of 0.0 and 1.0; and only the prompt template without counterexamples. We selected heroin and benzphetamine as our index terms for these follow-up experiments because these are both drugs of abuse that the DEA classifies as having higher and lower potential for abuse, respectively, than alprazolam. Additionally, we would expect heroin to be discussed at a rate similar to or higher than that of alprazolam, whereas we would expect much less discussion of benzphetamine.

#### 2.4.4. Google Filter

We used Google searches to automate an approximate validation of whether generated terms were synonymous with the index term. Upon extraction of each generated term from the GPT-3 response, we made a series of Google searches: the generated term alone, the generated term with “pill” appended, the generated term with “drug” appended, and the generated term with “slang” appended. The rationale behind the searches with appended keywords is that some drug slang terms have multiple meanings and a search of only the term itself may yield non-drug-related results; appending “drug”, “pill”, or “slang” makes it more likely to yield results with the drug-related context. We processed the search results through a specified maximum depth (e.g., a specified maximum depth of 10 would entail processing the top 10 search results), recording whether there is a search result within the maximum depth that has an instance of the index term appearing in its title or content snippet, and if so, the depth of the first result for which it does. Because the Google API limits the rate and daily number of API queries, we terminated the Google searching process for a term once one search contained a result with the index term. We also made the searching process more efficient with memoization.

#### 2.4.5. Drug Name Filter

We filtered out generated terms if they appeared in the set of RedMed index terms and were not the same as the queried index term. This choice was informed by the observation that GPT-3 tends to generate the name of different drugs with the same indications as the queried index term.

#### 2.4.6. Final Pipeline Parameters

After our parameter search experiments, we ran the final version of the pipeline on the set of 98 controlled index terms. We used a temperature of 1.0, a frequency penalty of 0.0, and a presence penalty of 0.0 for all GPT-3 queries. We used the Google filter with depth 10 and the drug name filter. We conducted 1000 iterations of the pipeline for each index term.

#### 2.4.7. Manual Labeling

To be able to evaluate pipeline performance, we manually labeled the terms generated by our pipeline for alprazolam and fentanyl. We chose to manually label these two index terms because these drugs are very widely abused and discussed and therefore of high interest for pharmacovigilance efforts; in addition to informing this study, generating gold standard labels for alprazolam and fentanyl may be useful for later pharmacovigilance research. For each unique generated term, a human labeler performs internet searches, searches directly on substance-related Reddit forums, and cross-references with compiled lists of known drug slang terms to determine if, by their best judgment, the generated term was a valid slang term, misspelling, brand name, or other synonym. All labelers had previous experience in drug-related informatics and were very familiar with the domain. We instructed labelers to mark a generated term as a synonym if they found at least one instance online of a person using that term in a context where it was apparent that they were referring to the index term or if it was the brand name of the index term in any country. This includes terms that have both drug meanings and non-drug meanings. For example, “bars” could refer to alprazolam, a long rod, an establishment serving alcoholic drinks, or the action of prohibiting something. Even though in many contexts, “bars” does not refer to alprazolam, it would be labeled as a synonym because there are contexts in which “bars” indisputably does refer to alprazolam. Terms in other languages were also accepted. For example, “alprazolan” is Spanish for alprazolam and is therefore a synonym of alprazolam. We acknowledge that it is possible that some manual labels may be incorrect, but given the expertise of the reviewers, we believe that such errors are scarce enough to not majorly impact the conclusions we draw from our results.

#### 2.4.8. Evaluation Criteria

We quantify the performance of our pipeline on the two index terms that we manually labeled by calculating precision (Equation (Equation 1)) and recall (Equation (Equation 2)). The F score is a metric to quantify the trade-off between precision and recall; the F1 score weights the two equally (Equation (Equation 3)), whereas the F2 score favors high recall over high precision (Equation (Equation 4)). We denote number of true positives by TP, number of false positives by FP, and number of false negatives by FN:(1)$$\mathrm{precision}=\frac{\mathrm{TP}}{\mathrm{TP}+\mathrm{FP}}$$
(2)$$\mathrm{recall}=\frac{\mathrm{TP}}{\mathrm{TP}+\mathrm{FN}}$$
(3)$$F1\;\mathrm{score}=2*\frac{\mathrm{precision}*\mathrm{recall}}{\mathrm{precision}+\mathrm{recall}}$$
(4)$$F2\;\mathrm{score}=5*\frac{\mathrm{precision}*\mathrm{recall}}{2*\mathrm{precision}+\mathrm{recall}}$$

In the context of this method, we prefer high recall to high precision when evaluating different filtering schemes. We do this because the primary use case for the lexicons produced by our pipeline is to scan social media posts for drug-related terms in order to identify which posts are likely about the drug of interest. In this context, it is better to flag irrelevant posts as relevant than to miss relevant posts because it is possible to use manual inspection or other automated models to further filter the posts, whereas the size of social media corpora makes it intractable to identify false negative posts. Therefore, we consider both the F1 and F2 scores in our evaluation.

We note that some applications of this lexicon or pipeline may require higher specificity or precision than provided by our current criteria, in which case subsequent filters will be needed to remove false positives. However, we maintain that favoring recall in an initial evaluation is important because, while it is possible to filter out likely false positives from an existing lexicon, it is much more difficult to introduce likely false negatives into an existing lexicon.

## 3. Results

### 3.1. Parameter Search

We first examined parameter trends from the results of the pipeline parameter sweep for the index term of alprazolam. We saw a clear relationship between increased temperature and increased number of UNGSes (Figure 2a). We also saw a large difference between the two prompt templates; the prompt template without counterexamples had dramatically more UNGSes than that with counterexamples (Figure 2b). There were not obvious relationships between frequency penalty or presence penalty and number of UNGSes, though for both penalties, we saw that the the maximum of the range of UNGSes per iteration tended to decrease as the penalties increased (Figure 2c,d). We found that the temperature trend was consistent when examining the results of the heroin and benzphetamine iterations (Figure 3a); we did not repeat the prompt variation as the alprazolam results were so stark. These two sets of iterations also showed a slight average decrease in number of UNGSes when either the frequency penalty or presence penalty was increased (Figure 3b,c). We therefore decided that a temperature of 1.0, a frequency penalty of 0.0, a presence penalty of 0.0, and the prompt template without counterexamples were the best parameter settings to use going forward. Further solidifying the decision to use these parameter settings, we observed that, for each drug, the set of 1000 iterations that generated the most UNGSes was one with all, or almost all, parameter settings matching our choices (Figure 4).

### 3.2. Google Search Depth Analysis

We used the manually labeled alprazolam and fentanyl data to both determine an appropriate cutoff for the maximum search depth and to examine if there is a relationship between search depth and manual label. We used a maximum search depth of 30 for the 1000 alprazolam iterations and saw a sharp decrease in the proportion of generated terms that are synonyms to generated terms that are not synonyms around a search depth of 10, in addition to an overall decrease in the number of unique terms generated (Figure 5a). This informed a lower maximum search depth of 10 for the 1000 fentanyl iterations, which displayed a similar power-law-like decrease in number of synonyms, non-synonyms, and unique terms generated overall as the depth increased (Figure 5b).

### 3.3. Generation Frequency Analysis

Across the 1000 iterations of the query pipeline, GPT-3 tended to generate many terms more than once. Notably, some very common colloquial names for the index terms used in these initial experiments appeared at a very high rate. We sought to investigate if a generated term’s frequency of generation could be used to estimate how likely it is to be a true synonymous term. For both the set of alprazolam iterations and the set of fentanyl iterations, we observed that terms generated only once or twice were overwhelmingly manually labeled as non-synonyms, and that most, but not all, terms generated more than 15 times were manually labeled as synonyms (Figure 6a,b). While it is possible that not all manual labels are correct, the trend still holds even if there are some erroneous labels. We examined the five most frequently generated terms for both alprazolam and fentanyl. The five most frequent alprazolam terms were “xanax”, “ativan”, “zoloft”, “alprazolan”, and “xanor”. Both “xanax” and “xanor” are common brand names of alprazolam, and “alprazolan” is both a common misspelling of alprazolam and the Spanish word for alprazolam. “Ativan” and “zoloft” are brand names of lorazepam and sertraline, respectively, which are distinct from alprazolam but share its anxiolytic effects. These two terms, which are not synonyms of alprazolam, were not caught by the drug name filter as they are brand names. The five most frequent fentanyl terms were “sublimaze”, “duragesic”, “fentanil”, “fentanylum”, and “fentora”, all of which are either brand names or common misspellings of fentanyl, and are therefore fentanyl synonyms.

After applying the Google search filter to the generated terms, we observed a reduction in the number of non-synonyms, most notably at the low end of the frequency range. Without the Google search filter, there were 137 alprazolam synonyms, 571 alprazolam non-synonyms, 168 fentanyl synonyms, and 907 fentanyl non-synonyms generated once (Figure 6a,b). With the Google search filter, there were 128 alprazolam synonyms, 115 alprazolam non-synonyms, 125 fentanyl synonyms, and 152 fentanyl non-synonyms generated once (Figure 6c,d). While, without the Google search filter, we may have discarded the terms only generated once or twice due to their high proportion of non-synonyms, we see that the proportion evens out after applying the Google search filter. We also note that, on the high-frequency end of the spectrum, the two non-synonyms present in the top five most frequently generated alprazolam terms do not pass the Google search filter, while the three synonyms do. Because of this effect, as well as the fact that there are synonyms at all frequency levels, we choose to not include a frequency-based filter into our query pipeline.

### 3.4. Pipeline Performance

We evaluated the performance of the lexicon generation pipeline using the manual labels for both alprazolam and fentanyl generated terms as a proxy for ground truth. In doing so, we sought to characterize the ability of each filter setup to automatically identify manually labeled synonyms and to determine which filters to run the pipeline with on a larger set of index terms.

As a baseline, we analyzed the performance when predicting that all terms generated by GPT-3 are synonyms (Figure 7a). This demonstrated how many synonyms were generated by GPT-3 for both alprazolam and fentanyl (269 and 314, respectively), but also showed how many non-synonyms are generated (750 and 1114, respectively). Despite the perfect recall in both cases (due to never assigning negative predicted labels), the low precision (0.264 and 0.220, respectively) supports our decision to filter the GPT-3 outputs.

We also analyzed the performance when only predicting a generated term as a synonym if it was already present in RedMed (Figure 7b). This led to perfect precision for both alprazolam and fentanyl (i.e., no false positives) but a low recall (0.178 and 0.115, respectively). The low recall is an indication of how many new terms that GPT-3 is generating that were not previously included in RedMed.

We analyzed multiple combinations of filters for the prediction of synonyms. The three filters assessed were the drug name filter, the frequency filter, and the Google search filter. The drug name filter removes generated terms that match any index term besides the queried index term. The frequency filter removes generated terms that are generated only once. We found that increasing the frequency threshold beyond one increased precision, but decreased recall; as previously stated, we prefer to maximize recall. The Google search filter removes generated terms if the corresponding index term does not appear in the first 10 Google search results for the term alone or with “pill”, “drug”, or “slang” appended. The precision, recall, and F1 and F2 scores for all filter combinations tested are shown in Table 2.

Because it generated both the highest F1 and F2 scores on the manual labels for both alprazolam and fentanyl, we used the classification scheme of the drug name filter and the Google search filter (but not using the frequency filter) to build the final lexicon for all drugs of abuse. We made this decision under the assumption that high F1 and F2 scores on the manual labels would correlate with high F1 and F2 scores on the (unknown) ground truth.

### 3.5. Drugs of Abuse Lexicon

We conducted 1000 iterations of the query pipeline on each the 98 index terms. On average, each index term had 3880 total generated terms and 1426 unique generated terms over the 1000 iterations, though this varies widely per drug (Figure 8a,b). All generated terms that passed the drug name filter and the Google search filter were compiled into a lexicon of GPT-3 synonyms for drugs of abuse. Each index term had an average of 141 unique GPT-3 synonyms in the lexicon (Figure 8c) and an average of 132 UNGSes (Figure 8d).

When only considering widely-discussed drugs, the observed distributions of the aforementioned counts shift. Widely-discussed drugs yielded more total generated terms on average (4063 per index term; Figure 8e) but fewer unique generated terms on average (1259 per index term; Figure 8f). They also yielded more unique GPT-3 synonyms on average (293 per index term; Figure 8g) and more UNGSes on average (268 per index term; Figure 8h).

We include Google search, drug name matching, and frequency information in the full lexicon to enable the addition or removal of filters in future applications.

## 4. Discussion

In this study, we demonstrate that GPT-3, a large language model trained on the entire internet and used extensively for few-shot text generation, is able to generate drug synonyms to facilitate pharmacovigilance based on social media. With automated API queries and simple automated filters, we create a lexicon of slang terms, misspellings, brand names, and other synonyms of drugs identified by the DEA as drugs of abuse with minimal manual intervention. We offer both the lexicon and the code used to create the lexicon for use in identifying drug-related social media posts and characterizing large-scale trends in drug abuse and overdoses.

Our lexicon allows researchers conducting pharmacovigilance on social media (or other text source that uses colloquial language without a controlled vocabulary) to easily scan a large amount of text data and flag posts that contain terms synonymous with a drug of interest. Not only is this approach very accessible, as it does not require the machine learning expertise or computational resources needed for advanced language models, but it also provides interpretability as it is clear which term is responsible for flagging each post. This interpretability can aid the removal of false positive examples. We hope that our lexicon enables pharmacovigilance to be more efficient and have lower latency, due to the ability to utilize social media data and the lack of a need to develop complicated machine learning models. Additionally, our pipeline can be used for easy synonym generation tasks in areas beyond pharmacovigilance.

While we found that GPT-3 generated hundreds of terms identified to be synonyms by manual labeling, the raw outputs also contained a large number of false positives, demonstrating the need for post-processing. We have shown that the drug name filter in combination with the Google filter yields the highest recall of all the filtering schemes. On average, our lexicon contains 141 GPT-3 synonyms per index term, and on average 132 of these are novel discoveries not found in RedMed. Importantly, these numbers increase for widely-discussed drugs that are more likely to be the focus of pharmacovigilance research. If we assume that the precision of the pipeline when generating fentanyl synonyms (the less precise of the two manually-labeled examples) holds for all index terms, then our lexicon contains 80 synonyms on average per index term, and 166 synonyms on average per widely-discussed index term. Notably, because GPT-3 is available as a pre-trained model, the process of querying GPT-3 and filtering the results to obtain these tens to hundreds of real synonyms requires relatively little effort, in direct contrast to RedMed’s word embedding model, which required its own training and tuning.

Our pipeline has some limitations. For example, our choice to prioritize high recall over high precision means that the resulting lexicon is likely to contain many false positives. The number of false positives may be additionally increased by our broad definition of positive examples in the manual labeling process (e.g., labeling “bars” as a positive example/synonym for alprazolam, when in many contexts it would be not be a synonym for alprazolam). If the application for which the lexicon is being used requires higher precision, then additional filters will need to be applied to remove false positives. Alternatively, one could generate a new lexicon using different pipeline parameters than those specified above.

GPT-3 is unlikely to be able to predict new drug slang. The version of GPT-3 that we use in our experiments completed training in late 2019. It therefore has no information about any event from 2020 and onward. While GPT-3 may produce plausible-sounding predictions of the future, it is important to remember that it is not an oracle and, unlike Google, does not have up-to-date access to the happenings of the world. Drug slang terms can shift with new media and trends in pop culture, and these shifts will not be represented in the outputs of GPT-3. Therefore, as time goes on, our pipeline may generate slang terms that become less relevant to the current state of online conversations about drug use. However, because GPT-3 is optimized for few-shot learning, it is possible to present it with recent knowledge and let it generate likely tokens from that. It is also not unlikely that OpenAI will release an updated GPT model in the future that will be trained on new internet content.

Similarly, we note that the use of the Google filter in the final version of the pipeline means that the generative capabilities of GPT-3 may be suppressed, in that a plausible novel slang term that is not yet in use online would be omitted from the final lexicon. This occurs because our current mode of pipeline evaluation depends upon online presence and would therefore also miss plausible novel terms. One may use our method without the Google search filter in an attempt to recover more such original terms, but they would need a different evaluation method or else risk an influx of false positives. However, we believe that this is not a major limitation, as the primary utility of this method in a pharmacovigilance context is that it can recover terms currently in use on the internet that may be unknown to pharmacovigilance researchers; the generation of a term that will never be used is not useful for monitoring trends in drug use.

We recognize that our method requires a set of existing synonyms (e.g., RedMed synonyms) to construct the initial prompts presented to GPT-3. In the absence of a relevant RedMed entry for a drug of interest, there are alternate ways that one can generate such a set of synonyms. First, one could use resources such as existing online slang term lists, specialized slang dictionary sites such as Urban Dictionary, social media sites such as Twitter and Reddit, or a simple Google search to manually gather a few example synonyms. Second, one could modify the GPT-3 prompt template to query GPT-3 for synonyms of the index term without providing examples, and manually validate the resulting terms through the aforementioned online resources. While either of these options would require extra manual processing, we believe that the amount of work required to obtain a few synonyms to construct a prompt pales in comparison to the amount of work saved when using our pipeline to generate novel synonyms based on that prompt, as the strength of GPT-3 is in few-shot learning.

Finally, our pipeline shares a common problem with many applications that use GPT-3: GPT-3 is so good at producing plausible outputs that it is very difficult to tell if an output is truth or fiction. Beyond our discussion of using automated filters to reduce the number of false positives, we must also address the philosophical question of whether it is appropriate to use GPT-3, or a similar generative language model, for this task at all. One could argue that the task might be better approached by training a new large language model to recognize drug-relevant text, or by using simpler AI methods than large language models. However, the incredible performance of GPT-3 across a range of text generation tasks, in addition to the evidence from our experiments with alprazolam and fentanyl terms, convinces us that there are enough synonyms produced by GPT-3 for it to be a valuable resource for social media pharmacovigilance. We encourage future users of our pipeline to carefully consider if GPT-3 is appropriate for their task of interest.

This work is primarily a proof-of-concept, and there are numerous improvements to the pipeline which could be made to further enrich the resulting lexicon. One such improvement would be additional prompt engineering to further maximize the number of synonyms generated per API query. For example, it is possible that changing the format of the synonym list from a numbered list to some other form may be beneficial. One could also consider other ways of giving GPT-3 examples of desired output (whether using the index drug or some other drug) to reduce generation of false positives. We acknowledge that the manual tuning of query parameters may have led to a suboptimal choice of parameter settings, though the fact that the three numerical query parameters were at their extremes (highest possible temperature and lowest possible nonnegative presence penalty and frequency penalty) suggests that this is unlikely to be the case. Nevertheless, future work could verify this choice with a parameter sweep conducted via an automatic optimization algorithm. Additionally, the release of ChatGPT [53], which was concurrent with the preparation of this manuscript, brings a newer model with more advanced capabilities to the research community. It is possible that using ChatGPT for this task instead of GPT-3 may yield better results.

We share these methods and results in the hopes of contributing to population-scale pharmacovigilance to combat the opioid epidemic and reduce harm from drug abuse. We do not condone the use of our lexicon or pipeline for censorship or surveillance at the individual level. We also acknowledge that our pipeline could be used to evade censorship or monitoring on online platforms, or could potentially otherwise influence the emergence of new slang. However, we believe that the chance of such influence is minor, and greatly outweighed by the potential for lexicons created from this pipeline to better inform understanding of large-scale trends in drug abuse.

## Figures and Tables

**Figure 1 biomolecules-13-00387-f001:**
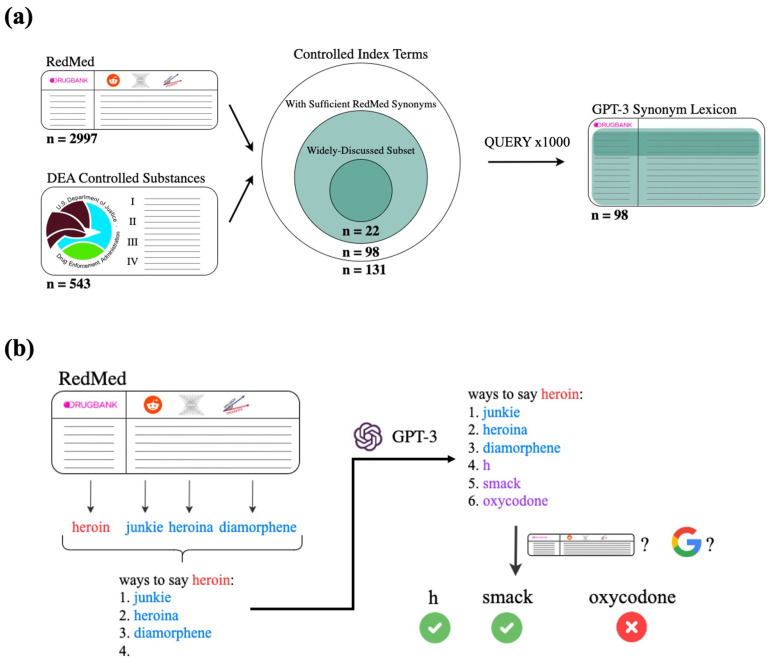
Overview of the selection steps and overall pipeline. (**a**) preprocessing. The intersection of RedMed and DEA controlled substances are taken as index terms to feed through the pipeline. Each index term goes through the pipeline for 1000 iterations, resulting in a lexicon of GPT-3 synonyms. Approximately one-quarter of the index terms put through the pipeline are designated as “widely-discussed” and are used to examine performance on terms with many synonyms and of high relevance to pharmacovigilance; (**b**) an example of a single iteration through the GPT-3 querying pipeline. For the desired index term (red), we uniformly sample three RedMed synonyms (blue) to insert into the prompt template. We present the prompt to the GPT-3 Completions API and parse the returned result for generated terms (purple). We use a Google search filter and a drug name filter to determine whether to classify generated terms as GPT-3 synonyms (green checkmark) or not (red x).

**Figure 2 biomolecules-13-00387-f002:**
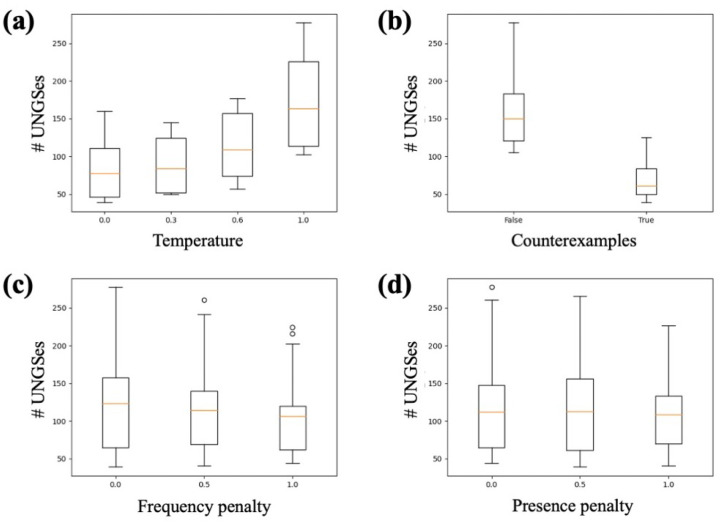
Parameter search using alprazolam as index term. Each combination of temperature, prompt template, frequency penalty, and presence penalty was used to conduct 1000 iterations of the query pipeline. The number of unique novel GPT-3 synonyms (UNGSes) generated by the 1000 iterations was recorded for each parameter set. Each subfigure shows the distribution of UNGSes for each value of (**a**) temperature; (**b**) prompt template; (**c**) frequency penalty; and (**d**) presence penalty.

**Figure 3 biomolecules-13-00387-f003:**
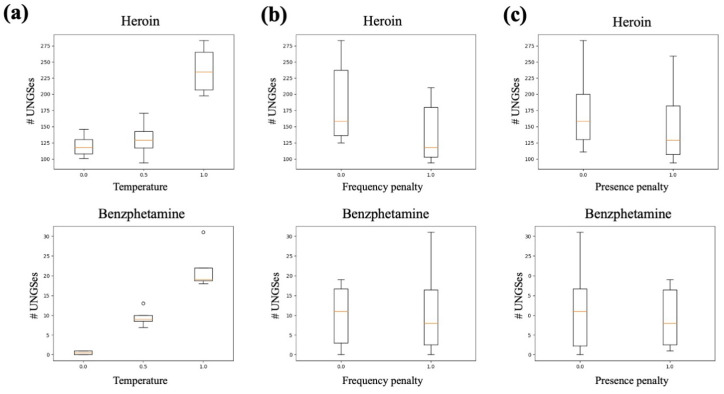
Smaller parameter search using heroin and benzphetamine as index terms. Each combination of temperature, frequency penalty, and presence penalty was used to conduct 1000 iterations of the query pipeline. The number of UNGSes generated by the 1000 iterations was recorded for each index term and parameter set. Each subfigure shows the distribution of UNGSes for both heroin and benzphetamine for each value of (**a**) temperature; (**b**) frequency penalty; and (**c**) presence penalty.

**Figure 4 biomolecules-13-00387-f004:**
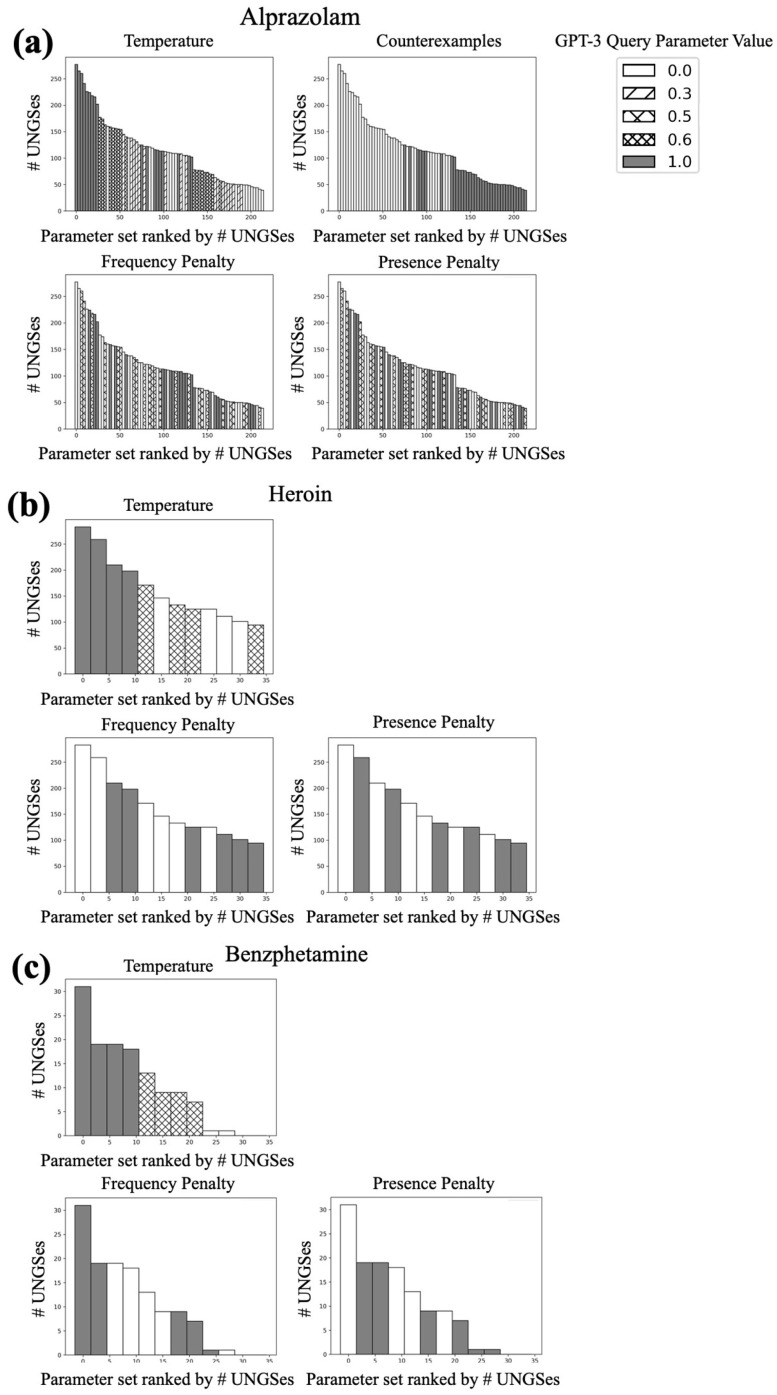
Number of UNGSes per each combination of temperature, prompt template, frequency penalty, and presence penalty for (**a**) alprazolam; (**b**) heroin; and (**c**) benzphetamine. The bar shading in each subplot represents the value of the parameter indicated in the title of that subplot. Each bar represents a different parameter set used for 1000 iterations of the pipeline.

**Figure 5 biomolecules-13-00387-f005:**
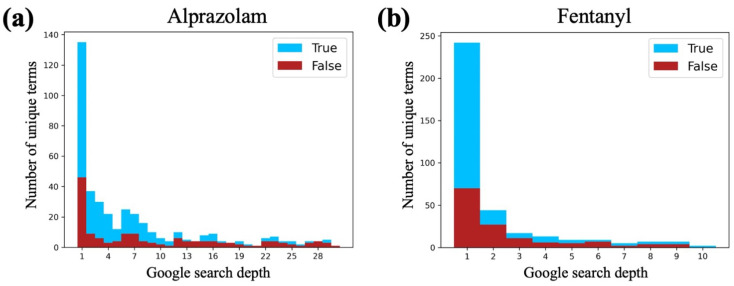
Histogram of the number of unique terms generated at each depth in the Google search for both (**a**) alprazolam and (**b**) fentanyl. At each search depth, the count of synonyms (true examples) is shown as blue bars, and the count of non-synonyms (false examples) is shown as red bars. The blue bars are stacked on top of the red bars (i.e., they do not continue behind the red bars). The alprazolam queries allowed a maximum search depth of 30, whereas the fentanyl queries were limited to a maximum search depth of 10 as utility drops after the tenth result.

**Figure 6 biomolecules-13-00387-f006:**
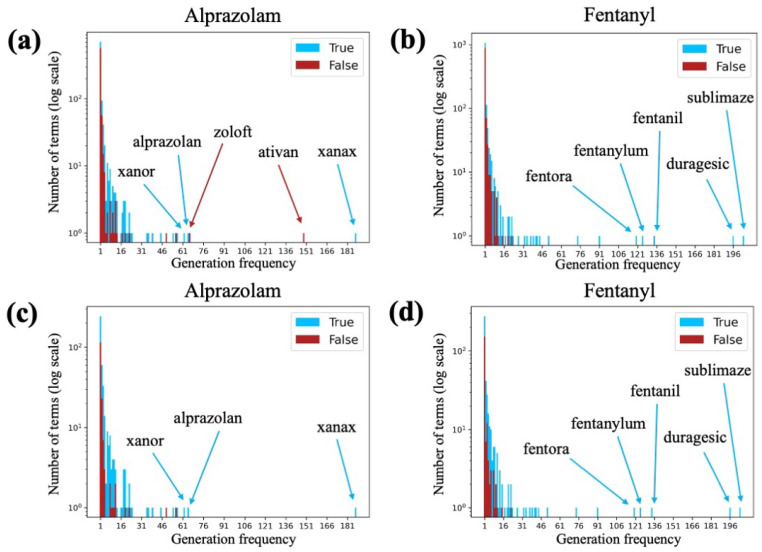
Histogram of the number of terms generated at different frequencies for (**a**) alprazolam without the Google search filter; (**b**) fentanyl without the Google search filter; (**c**) alprazolam with the Google search filter; and (**d**) fentanyl with the Google search filter. Plots use a logarithmic scale. At each search depth, the count of synonyms (true examples) is shown as blue bars, and the count of non-synonyms (false examples) is shown as red bars. The blue bars are stacked on top of the red bars (i.e., they do not continue behind the red bars). The top five most generated terms for each drug are labeled. All plots omit generated terms that do not pass the drug name filter.

**Figure 7 biomolecules-13-00387-f007:**
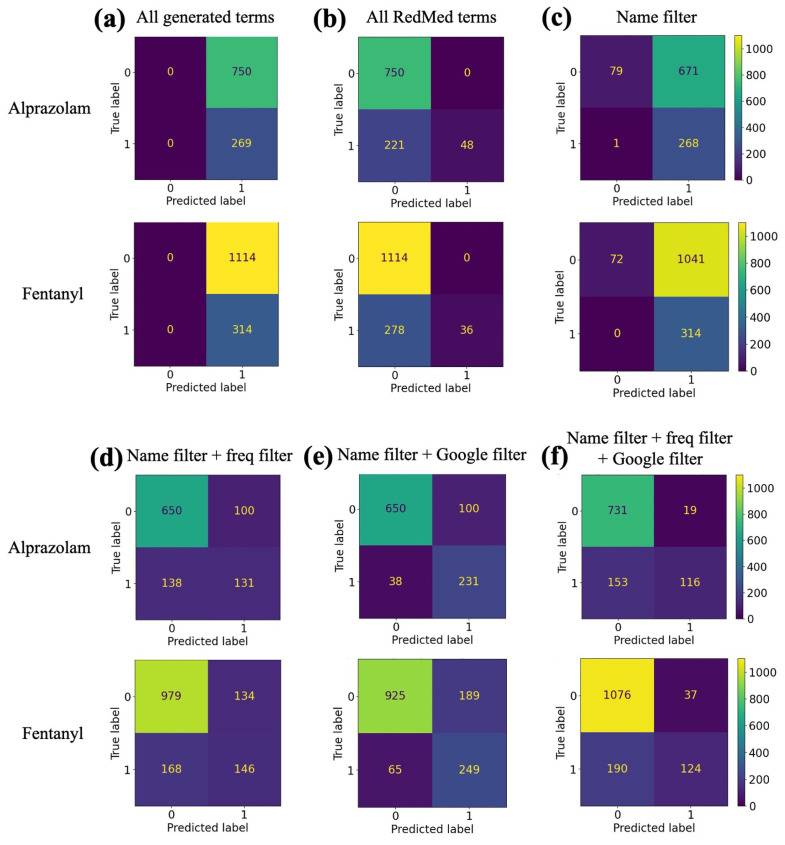
Confusion matrices for both alprazolam and fentanyl queries. True labels are determined by manual labeling. Predicted labels are determined by (**a**) classifying all generated terms as true; (**b**) classifying all generated terms that appear in RedMed as true; (**c**) classifying all generated terms that pass the drug name filter as true; (**d**) classifying all generated terms that pass the drug name filter and the generation frequency filter as true; (**e**) classifying all generated terms that pass the drug name filter and the Google search filter as true; and (**f**) classifying all generated terms that pass the drug name filter, the generation frequency filter, and the Google search filter as true. In each confusion matrix, a 0 denotes a negative classification, which is a non-synonym, and a 1 denotes a positive classification, which is a synonym.

**Figure 8 biomolecules-13-00387-f008:**
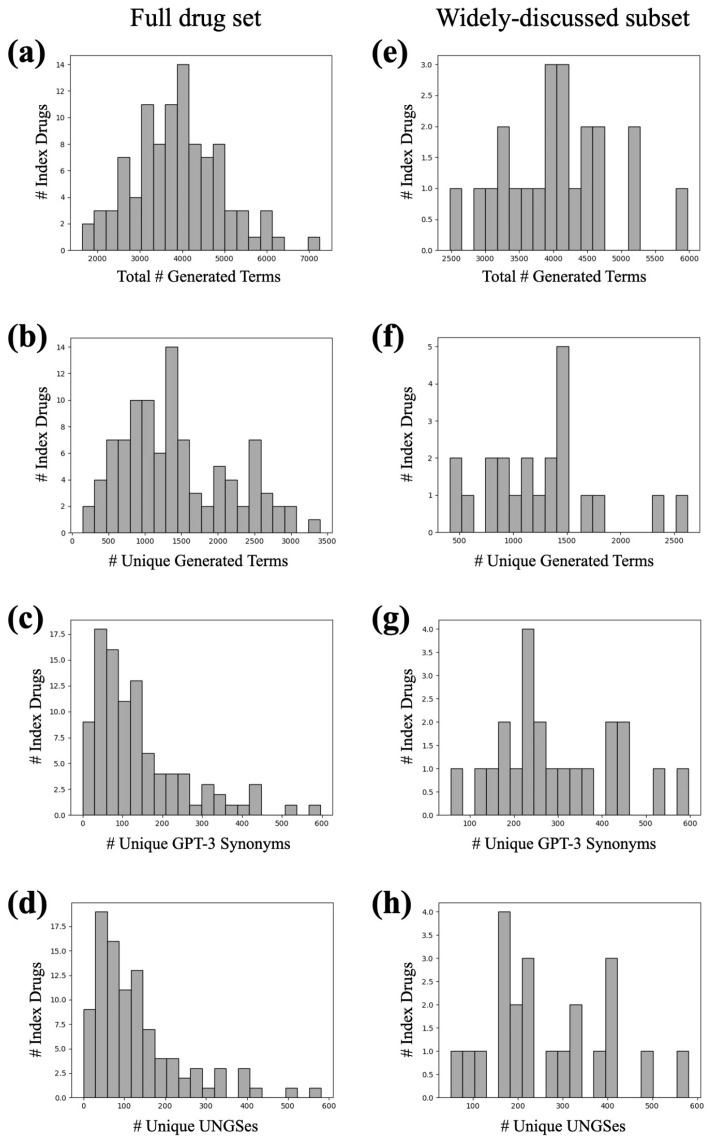
Histograms showing various distributions of quantities for each index term in the full lexicon of 98 drugs of abuse (**a**–**d**) and the subset of 22 widely-discussed drugs of abuse (**e**–**g**). Quantities depicted are total generated terms (**a**,**e**), unique generated terms (**b**,**f**), unique GPT-3 synonyms (generated terms passing filters) (**c**,**g**), and UNGSes (generated terms passing filters and not present in RedMed) (**d**,**h**).

**Table 1 biomolecules-13-00387-t001:** Definitions of new terminology used throughout this manuscript.

Term	Abbreviation	Definition
*Associated known drug term*	AKDT	As defined in [40], known terms used synonymously for a given drug, most often brand names.
*Controlled index term*	-	A *controlled substance* that is an *index term*.
*Controlled substance*	-	A substance (i.e., drug) that is deemed to have a high potential for abuse by the DEA and is therefore controlled.
*Generated term*	-	See *GPT-3 generated term*.
*GPT-3 generated term*	-	A term generated from a GPT-3 query as a candidate synonym for the corresponding index term used in the prompt.
*GPT-3 synonym*	-	A *GPT-3 generated term* that has been automatically labeled as a synonym following a filtering scheme.
*Index term*	-	The identifying term of a drug as indexed in RedMed; also the generic name of a drug as indicated in DrugBank.
*Novel GPT-3 synonym*	-	A *GPT-3 synonym* that is not already present in RedMed as a *RedMed synonym*.
*Non-synonym*	-	A *generated term* that has been manually labeled as not synonymous for the corresponding queried *index term*.
*RedMed synonym*	-	A term listed in RedMed as synonymous for a given *index term*.
*Synonym*	-	A *generated term* that has been manually labeled as synonymous for the corresponding queried *index term*.
*Unique novel GPT-3 synonym*	UNGS	Equivalent to a *novel GPT-3 synonym* but specifying that each unique *novel GPT-3 synonym* is only counted once no matter how many times it has been generated.
*Widely-discussed*	-	Specifying that a drug appears relatively more frequently on Reddit, suggesting higher rates of online discussion, more synonymous terms, and potentially greater interest for pharmacovigilance.

**Table 2 biomolecules-13-00387-t002:** Evaluation metrics when using different classification schemes for GPT-3 synonyms and using manual labels as a proxy for ground truth.

Index Term	GPT-3 Synonym Criteria	Precision	Recall	F1 Score	F2 Score
Alprazolam	All generated terms	0.264	1.000	0.418	0.642
Fentanyl	All generated terms	0.220	1.000	0.361	0.585
Alprazolam	All RedMed terms	1.000	0.178	0.302	0.213
Fentanyl	All RedMed terms	1.000	0.115	0.206	0.140
Alprazolam	Drug name filter	0.285	0.996	0.443	0.664
Fentanyl	Drug name filter	0.232	1.000	0.377	0.602
Alprazolam	Drug name & frequency filters	0.567	0.487	0.524	0.501
Fentanyl	Drug name & frequency filters	0.521	0.465	0.491	0.475
Alprazolam	Drug name & Google filters	0.698	0.859	0.770	0.821
Fentanyl	Drug name & Google filters	0.568	0.793	0.662	0.735
Alprazolam	Drug name, frequency, & Google filters	0.859	0.431	0.574	0.479
Fentanyl	Drug name, frequency, & Google filters	0.770	0.395	0.522	0.438

## Data Availability

The drugs of abuse lexicon, accompanying filter data, and pipeline code are all available at https://github.com/kristycarp/gpt3-lexicon.

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
