# Peer review of "Using GPT-3 to Build a Lexicon of Drugs of Abuse Synonyms for Social Media Pharmacovigilance"

_biomolecules, 2023, doi:10.3390/biom13020387_

Round 1
Reviewer 1 Report
In this study, authors used GPT-3 to analyze social media text for the purpose of pharmacovigilance to solve the problem of drug abuse. The study argues and tries to prove that GPT-3 is valuable for social media pharmacovigilance. The authors made the data and the code available. I have the following comments:
1. At the end of the Introduction Section, a list of well-defined contributions must be added.
2. All abbreviations should be defined in their first use.
3. How did you define the “Widely-discussed drugs of abuse”?
4. A figure for the full pipeline is needed to show the full picture of the proposal.
5. The parameters tuning has been done manually. How can you assure that these are the optimum parameters? The same for Google Search Depth Analysis, why did not use an automatic search optimization algorithm.
6. Did you check different formats for the queries?
7. “… For both the set of alprazolam iterations and the set of fentanyl iterations, we observed that terms generated only once or twice were overwhelmingly manually labeled as non-synonyms, and that most, but not all, terms generated more than 15 times were manually labeled as synonym…” Is this sufficient to know that a term is synonym? How you consulted domain experts?
8. In the “Pipeline performance”, what is the benchmark performance that you are comparing with. How can we know that these are the best results. How can we know that a term is indeed a synonym for a drug? You are using three filters to determine, but who said that the result after the three filters is correct?
9. Authors are required to share the results of the lexicon of GPT-3 synonyms for drugs of abuse.
Reviewer 2 Report
Please see attached document for comments and suggestions.

Round 2
Reviewer 1 Report
Accept.
Thank you.